# Temporal Evaluation of Insecticide Resistance in Populations of the Major Arboviral Vector *Aedes Aegypti* from Northern Nigeria

**DOI:** 10.3390/insects13020187

**Published:** 2022-02-10

**Authors:** Muhammad M. Mukhtar, Sulaiman S. Ibrahim

**Affiliations:** Department of Biochemistry, Bayero University, Kano PMB 3011, Nigeria; muhammadmahemukhtar@gmail.com

**Keywords:** *Aedes aegypti*, Nigeria, insecticides, temporal, increase, resistance, metabolic, enzymes, *kdr*

## Abstract

**Simple Summary:**

Outbreaks of dengue and yellow fever are fast becoming normal in Nigeria. Transmitted by mosquito vectors, *Aedes aegypti* and/or *Aedes albopictus*, control of these arboviral diseases depends largely on control of the above vectors. This requires knowledge of the identity/composition and insecticide resistance profile of the vector species—knowledge which is inadequate across Nigeria. In this study we characterised two populations of *Aedes aegypti* from north-western Nigeria (BUK/Kano and Pantami). Bioassays with Kano larvae suggest resistance to temephos (an important insecticide for larval control) is low but increasing, while deltamethrin resistance is high, and increased 6-fold between 2018 and 2019, and 11-fold by 2020. Adult bioassays established high pyrethroid resistance (the bed net insecticides) and extreme resistance to DDT (an indoor residual spraying insecticide). Bioassays with piperonylbutoxide and diethylmaleate (synergists which block activity of the enzymes that confer insecticide resistance) resulted in significant recovery of mortalities, implicating CYP450s and GSTs—enzymes which confer pyrethroid and DDT resistance, respectively. Tests with pyrethroid-containing bed net, PermaNet^®^ 3.0 (side panels) revealed high resistance, in contrast to the areas also containing piperonylbutoxide (PermaNet^®^ 3.0, roof panel). These findings highlight the challenges associated with the control of this arboviral vector of public health importance in Nigeria.

**Abstract:**

To support evidence-based control measures, two Nigerian *Aedes* populations (BUK and Pantami) were characterised. Larval bioassay using temephos and deltamethrin revealed a significant increase in deltamethrin resistance, with LC_50_ of 0.018mg/L (resistance ratio compared to New Orleans, RR = 2.250) in 2018 increasing ~6-fold, by 2019 (LC_50_ = 0.100mg/L, RR = 12.5), and ~11-fold in 2020 (LC_50_ = 0.198mg/L, RR = 24.750). For the median deltamethrin concentration (0.05mg/L), a gradual decrease in mortality was observed, from 50.6% in 2018, to 44.9% in 2019, and 34.2% in 2020. Extremely high DDT resistance was observed, with <3% mortalities and LT_50_s of 352.87 min, 369.19 min and 406.94 min in 2018, 2019 and 2020, respectively. Significant temporal increase in resistance was observed towards ƛ-cyhalothrin (a type II pyrethroid) over three years. Synergist bioassays with diethylmaleate and piperonylbutoxide significantly recovered DDT and ƛ-cyhalothrin susceptibility respectively, implicating glutathione S-transferases and CYP450s. Cone bioassays revealed increased resistance to the PermaNet^®^ 3.0, side panels (mortalities of 94% in 2018, 66.4% in 2019, and 73.6% in 2020), while full susceptibility was obtained with the roof of PermaNet^®^ 3.0. The F1534C *kdr* mutation occurred in low frequency, with significant correlation between heterozygote genotypes and DDT resistance. This temporal increase in resistance is a major challenge for control of this vector of public health importance.

## 1. Introduction

The arboviral vector *Aedes aegypti* is the most invasive species of mosquitoes, now present in nearly every tropical and sub-tropical region of the world, posing a threat to health globally [1,2,3,4]. It is responsible for transmitting most of the arthropod-borne viruses (arboviruses) of public health importance, including dengue virus, Zika, chikungunya, and yellow fever viruses, and sometimes filarial worm [2,3,4]. Different types of arboviral diseases are transmitted by this urban mosquito affecting more than 120 countries, many of which are low-income and middle-income, with ruinous effect on human health and economic development [4]. The Flaviviruses are responsible for haemorrhagic and encephalitic diseases predominantly vectored by *Ae. aegypti* and *Ae. albopictus*, as well as the *Culex* species [5,6,7]. The clinically-important mosquito-borne flaviviruses comprise many important human pathogens, transmitted by the Aedes spp, such as Dengue virus (DENV), Japanese encephalitis virus (JEV), West Nile virus (WNV), yellow fever virus (YFV) and Zika virus (ZIKV) [8]. The World Health Organisation [9] has ranked dengue as the most salient mosquito-borne viral disease worldwide.

The emergence and spread of all four serotypes of DENV (DENV1-4) across the regions of the world pose a great threat of a global pandemic [7,10]. The worldwide prevalence rate of dengue has grown vastly in recent decades with about half of the world’s population presently at risk of infection [11,12,13]. There are an estimated 100–400 million dengue infections each year, though the majority of cases are asymptomatic or mild, and easy to manage, which makes it difficult to report the accurate incidence [12]. However, Bhatt and colleagues [13] have used a modelling framework to estimate the global dengue virus infections per annum as 390 million cases (95% credible interval 284–528 million). The world experienced the worst devastating outbreak of dengue fever in the year 2016 with the Americas region reporting more than 2.38 million cases, and Brazil alone contributing about 1.5 million cases, with over 1000 deaths, which tripled the Brazilian incidence of the year 2014 [13]. The prevalence of DENV infection is high in the African continent, with substantial heterogeneity between different regions [14,15,16,17], and the number of sporadic and epidemic dengue fever cases increasing in recent years in West African countries [17].

Six viruses causing haemorrhagic fever, including the dengue virus, yellow fever virus, and Rift valley fever virus, have been isolated from Nigeria [18]. However, only four of these have caused haemorrhagic disease among Nigerians. Adekolu-John and Fagbami [19] have reported the presence of dengue virus in Kainji Lake area of Nigeria in the 1980s. From then onward, several researchers have documented the existence of dengue virus in various parts of the country, including northern Nigeria. Dengue virus antibodies (anti-DENV IgG) have been reported in the Guinea Savannah region of Nigeria with prevalence of 20.5% [20]. Likewise in Kano (north-western Nigeria) high prevalence of dengue IgM has been reported from patients reporting with febrile fever [21].

The worldwide prevalence rate of yellow fever virus (YFV) was estimated as 200,000 cases, with mortality of about 30,000 per annum, 90% of which occur in Africa [22]. For over 60 years Nigeria has been experiencing a sporadic outbreak of YFV diseases, especially in the southern part of the country where the YFV was first isolated [23,24,25]. Recently, outbreaks of YFV diseases have been reported in more than half of the local government nationwide, with an on-going infection in Bauchi, Benue, Delta, Ebonyi, and Enugu states [22]. During the first 8 months of 2021 alone, there were a total of 1,312 confirmed cases of YFV diseases in Nigeria [25].

The highest number of ZIKV infection was reported from the United States (81,115), followed by Thailand (1044), Mexico (996) and India (671) during the period of 2016 to 2019 [26]. Zika virus infection during pregnancy is a cause of microcephaly and other congenital abnormalities in the developing foetus and new-born, in addition to pregnancy complications such as foetal loss, stillbirth, and preterm birth (https://www.who.int/news-room/fact-sheets/detail/zika-virus, accessed on 1 December 2021).

Chikungunya virus (CHKV) was first isolated and characterized in 1953 during an epidemic of febrile polyarthritis in Tanzania [27]. From then onward, several findings have shown the serological evidence of ZIKV infections, including in Nigeria [28,29,30].

Resistance to all the classes of insecticides by *Ae. aegypti* and *Ae. albopictus* and the contribution of knockdown resistance (*kdr*) mutation in the voltage-gated sodium channel (VGSC) in insecticide resistance have been reported in many parts of the world. From Asia, Ishak and colleagues [31] have reported moderate resistance to temephos across Malaysia and malathion only in the central region of the country (Kuala Lumpur). Overexpression of CYP450 genes play critical roles in resistant to pyrethroid and DDT in *Ae. aegypti* [32] with several CYPP450s (*CYP9J27, CYP6CB1, CYP9J26* and *CYP9M4*) found as the most overexpressed in the resistant *Ae. aegypti*. Several GSTs (e.g., *AaGSTD1* and *AaGSTE2*) were also reported to be overexpressed in pyrethroid and DDT resistant *Ae. aegypti* [33]. Carboxylesterases (CCEs) have been generally associated with organophosphate (OP) resistance in Aedes spp, though they likely play important roles in pyrethroid resistance as well [34,35]. Many studies have also reported increased activities of all the three major detoxification enzymes (CYP450s, GST and CCEs) (biochemically) in resistant Aedes mosquitoes which further prove their role in resistance [36,37,38,39]. Synergist bioassays were also observed to reverse the *Ae. aegypti’*s resistance to the DDT, pyrethroids and carbamate [32,39,40]. Primarily, the VGSC mutations F1534C, V1016G, V1016I, V410L, G923V, L982W, S989P, I1011M, I1011V, T1520I, and D1763Y mediate the *kdr* resistance in *Aedes stegomyia*, to all the four classes of insecticides (pyrethroids, DDT, carbamates and organophosphates), but only F1534C, V1016G, I1011M, V410L have been validated as being directly linked to insecticide resistance [40,41]. Significant associations exist between pyrethroid and DDT resistance in the Malaysian populations of *Ae. aegypti* and the presence of F1534C mutation, but not with V1016G. However, the resistance increases additively when the two *kdr* mutations exist concurrently [32]. High pyrethroid resistance but susceptibility to organophosphates and bendiocarb has been reported in Indonesian *Ae. aegypti*, where the resistance correlates with the presence of V1016G mutation [42,43]. Recently, Yougang and colleagues [44] reported for the first time F1534C mutation and contrasting patterns of resistance to all the four classes of insecticides in Cameroon and implicated the F1534C mutation in resistance to the pyrethroids and DDT. Previous reports from Cameroon, Burkina Faso and Congo also revealed different patterns of resistance to pyrethroids and DDT, with partial restoration of resistance after pre-exposure to synergists, diethylmaleate and piperonylbutoxide [44,45,46,47,48].

Records of the insecticide resistance profile of *Ae. aegypti* from Nigeria and molecular mechanisms is scanty. Moderate resistance to DDT and marginal resistance to deltamethrin have been reported in *Ae. aegypti* populations from Lagos (south-western Nigeria) [49]. Another study from south-eastern Nigeria (Anambra) revealed high resistance to DDT, marginal resistance to propoxur, but complete susceptibility to deltamethrin and pirimiphos-methyl; while *Ae. albopictus* shows a different pattern of resistance, with moderate resistance to DDT, slight tolerance to deltamethrin and pirimiphos-methyl, and a total susceptibility to propoxur [50]. Control of the arboviruses depends largely on vector control using insecticides. However, with resistance in Aedes mosquitoes increasingly reported across Africa [51] and more than 20 laboratory confirmed epidemics in more than 20 African countries, between 1960 and 2017 [16], it is necessary for the African countries to scale up efforts to prevent and/or control outbreaks. However, this requires primary data on resistance profiles and mechanisms in the Aedes populations, which is lacking, for example from northern Nigeria, slowing evidence-based control measures and resistance management.

In the present study, the insecticide resistance profile of *Ae. aegypti* mosquitoes from northern Nigeria was characterised, establishing multiple resistance and intensely high resistance towards DDT, as well as resistance to a conventional LLIN, PermatNet^®^ 3.0 (side panels). Temporal increase in resistance to ƛ-cyhalothrin (a type II pyrethroid) was observed over three consecutive years. The possible role of metabolic mechanisms driving the resistance in the field was established through synergist bioassays (with significant recovery of susceptibility towards DDT and ƛ-cyhalothrin). Genotyping detected the voltage-gated sodium channel F1534C *kdr* mutation in low frequency and associated with DDT resistance in heterozygote (FC) females.

## 2. Materials and Methods

### 2.1. Mosquito Sampling and Rearing

The Aedes larvae and pupae were collected from the breeding containers and stagnant water [52] using ladles and pipets. Black buckets and tyres containing water were also used, from which eggs and larvae were collected within 48–72 h. Collection was carried out in Bayero University, Kano, BUK (11°58′45.4″ N, 8°28′47.9″ E), Nigeria, between July to October in 2018 and 2019, and August to November 2020. For Pantami town (10°16′00.8″ N, 11°09′57.4″ E), Gombe, Nigeria collection was conducted in August 2020 only (and mosquitoes used for adult bioassays only). Mosquitoes were reared in an insectary, in the Biochemistry Department, BUK. Larvae were separated based on stages of development (1st to 4th instar) into shallow, enamel plastic trays at a density of about 100 larvae/L of deionized water or the water collected from breeding sites. Larvae were maintained on chinchilla pellets, supplemented with brewer’s yeast tablets. The emerged adults were fed 10% sucrose, and maintained at standard insectary conditions of temperature 27 °C ± 2, relative humidity of 75% ± 10 and a 12:12 h light-dark cycle [53].

### 2.2. Mosquito Populations Identification to the Species Level

A total of 880 adults *Aedes* from those subjected to WHO tube bioassays were identified using morphological keys [54]. These include 800 from BUK (200 from 2018 collection, 300 each from 2019 and 2020 collections) and 80 from 2020 collection in Pantami. The pictorial keys were utilised, using a 7X–45X, Trinocular XTL Stereo Zoom microscope to identify the Aedes species. Two individuals examined the mosquitoes independently.

### 2.3. Insecticide Susceptibility Profiling of Aedes Populations

#### 2.3.1. Larval Bioassays

Larval bioassays were carried out to investigate temephos and deltamethrin resistance and temporal changes in the resistance. This was conducted using larvae collected for three consecutive years (2018, 2019 and 2020) in Kano, and according to the protocol of WHO [55]. From each year, a total of 540 4th instar larvae (L4) were used for the experiment. A total of 1 mL of 1.0 mg/mL of temephos (Merck, Darmstadt, Germany) was diluted in 10 mL of absolute ethanol from which 1 mL was mixed with 99 mL of ddH_2_O into small plastic cups, to produce a concentration of 40 part per million (ppm)/mg/L. This stock solution was serially diluted 7× (40–0.00004 ppm) to span the diagnostic concentration of 0.012 ppm recommended by the WHO [55]. Similar test concentrations were prepared for the type II pyrethroid, deltamethrin (Merck, Darmstadt, Germany). Four replicates of 20 L4 larvae were introduced to each test cup for all the seven concentrations. Mortalities were scored 24 h post-exposure. The lethal concentrations that kill 50% (LC_50_) and 90% (LC_90_) of the larvae were estimated using probit analysis. The lethal concentrations for temephos and deltamethrin were compared with the LC_50s_ previously established for the fully susceptible laboratory colony, the New Orleans, for deltamethrin [56] and temephos [57]. These allowed for calculations of the resistance ratios (RR).

#### 2.3.2. Adult Insecticide Bioassays

Adult bioassays were conducted following the protocol of WHO for Aedes mosquitoes [58]. For the Kano population the insecticides were pyrethroids: 0.25% and 0.75% permethrin, 0.03% and 0.05% of deltamethrin, ƛ-cyhalothrin, and α-cypermethrin, and 0.15% cyfluthrin; organochlorides: 4% of DDT and dieldrin; organophosphate: 1% fenitrothion; and carbamate: 0.1% propoxur. Minimum of four replicates each of 25 *Aedes aegypti* females (3–5 day old) were exposed for 1 h. Mosquitoes were transferred to holding tubes and fed with sugar for 24 h before mortalities were recorded, and dead females stored in silica gel (Fisher Scientific, Loughborough, UK). For control, during each experiment, two replicates of 20–25 females from the same populations, and of the same age were kept in holding tubes without insecticide exposure. For the Pantami collection of 2020 only 4% DDT was tested.

#### 2.3.3. Time-Course Bioassays 

To investigate resistance intensity with time [59,60], additional bioassays were conducted with the 2018, 2019 and 2020 BUK population of *Ae. aegypti* by varying the exposure time with the discriminating concentration of DDT. Four replicates of 20–25 female *Ae. aegypti* (3–5 d old), were exposed to 4% DDT for 1 h, 2 h, 3 h or 9 h, to determine the lethal time at which 50% and 90% of the mosquitoes die (LT_50_ and LT_90_ respectively). Bioassays were conducted as outlined above and mortalities were recorded at 24 h.

#### 2.3.4. Investigation of the Role of Metabolic Resistance Enzymes Using Synergist Bioassays

Two synergists, 4% piperonylbutoxide (PBO), an inhibitor of cytochrome P450 monooxygenases [61], and 8% diethylmaleate (DEM), an inhibitor of glutathione S-transferases (GSTs) [62], were used to assess the role of metabolic enzymes in resistance to the ƛ-cyhalothrin (PBO) and DDT (DEM). For each synergist, four replicates of 20–25 (2–5 d old) females collected from BUK in 2018, 2019 and 2020 were first pre-exposed to the synergist-impregnated papers for 1 h before they were transferred to tubes containing 0.05% ƛ-cyhalothrin for PBO or 4% DDT for DEM. Following exposure for 1 h mosquitoes were transferred to holding tubes, supplied with 10% sugar and mortalities recorded after 24 h. For each insecticide/synergist experiment two controls were set: (i) 2 replicates of 20–25 females exposed to synergist only; and (ii) 2 replicates of 20F25 females exposed to insecticides only. Mortalities were scored 24 h post exposure.

### 2.4. Determination of Efficacy of a LLIN (PermaNet^®^ 3.0)

Cone bioassays [63] were conducted to assess the efficacy of Permanent^®^ 3.0. Previous studies have encouraged the use of insecticide treated materials, such as the LLINs for control of diurnally active *Ae. aegypti* [64,65]. In the course of our Anopheles collection in Kano, we have found on several occasions some blood fed Aedes mosquitoes resting indoors. The side panels containing 2.1–2.8 g/kg ± 25% deltamethrin and the roof (4.0 g/kg ± 25% deltamethrin, combined with 25 g/kg ± 25% of PBO) were used. Fifty replicates each of five females *Ae. aegypti* from Kano (3–5 day old) were exposed to each piece of netting (25 cm × 25 cm, randomly cut) for 3 min under standard WHO cones, after which they were transferred to paper cups and held for 24 h with access to 10% sucrose solution. The knock-down and mortality were recorded after 1 h and 24 h post-exposure, respectively. For controls, 10 replicates of 5 mosquitoes were also exposed to untreated nets under the same condition.

### 2.5. Investigation of the Role of VGSC F1534C and V1016G Knockdown kdr Mutations in ƛ-Cyhalothrin and DDT Resistance

A total of 155 mosquitoes from BUK were genotyped for the F1534C and V1016G *kdr* mutations. This comprises 42 ƛ-cyhalothrin-alive and 42 -dead, as well as 34 of DDT-alive and 37 of DDT-dead. The F1534C allele specific PCR (AS-PCR) was performed as previously established [66]. The PCR reaction mix of 15 μL contained 1 µL each of the gDNA, 1.5 µL of 10x Taq Buffer A, 0.4 mM (0.5 µL) of each of forward and reverse primers, 1.25 mM (0.75 µL) of MgCl_2_, 0.84 mM (0.5 µL) of dNTP mixes and 0.2 µL of Kapa Taq DNA polymerase (KAPA Biosystems, Wilmington, MA, USA), in ddH_2_O. Four primers were used, of which AaEx31P: 5’-TCG CGG GAG GTA AGTT ATTG-3’ and AaEx31Q: 5’-GTT GAT GTG CGA TGGA AATG-3’ amplified a control band of 350 bp, while two internal allele-specific primers AaEx31wt: 5’-CCT CTAC TTTG TGTT CTTC ATCA TCTT-3’ and AaEx31mut: 5’-GCG TGAA GAAC GACC CGC-3’ produced 231 bp products (“wild-type” phenylalanine allele) and 167 bp (“mutant” cysteine allele) respectively, with heterozygote individuals producing the above two bands. Amplification was carried out using the following conditions: initial denaturation of 2 min at 94 °C, followed by 30 cycles each of 30 s at 94 °C(denaturation), 30 s at 55 °C (primer annealing) and 30 s at 72 °C (extension). This was followed with 10 min final extension at 72 °C. The PCR products were separated on a 2% agarose gel stained with pEqGREEN.

The allele-specific genotyping for V1016G was also carried out as previously [67] using the same amount of reaction mix and thermocycling conditions, as above. The primers used for the genotyping were V1016G_F: 5’-ACC GAC AAA TTG TTT CCC-3’, V1016G_Val-R: 5’-GCG GGC AGC AAG GCT AAG AAA AGG TTA ATTA-3’ and V1016G_Gly-R: 5’-GCG GGC AGG GCG GCG GGG GCG GGG CCAGC AAG GCT AAG AAA AGG TTA ATTA-3’. The amplified products were separated on 3% agarose gel, stained with pEqGREEN. Alleles were discriminated based on sizes of bands on gels, with 80 bp fragments indicating homozygote resistant allele (GTA for valine to GGA for glycine); 60 bp representing homozygote susceptible allele (no mutation); while the heterozygote alleles presented both bands.

### 2.6. Data Analysis

Linear probit analysis of larval bioassay results for LC_50_ and for the adult quantitative bioassay for LT_50_ was performed using PASW statistics 18 software (http://www.spss.com.hk/statistics/, accessed on 1 December 2021). The resistance ratio was calculated by dividing the calculated LC_50_ by the LC_50_ from the fully susceptible laboratory population, New Orleans. Analysis of WHO tube bioassay results and plots were performed using Microsoft excel 2016. For synergist bioassays, Chi square test of significance was performed using an online tool (https://www.socscistatistics.com/tests/chisquare/default2.aspx, accessed on 1 December 2021). The correlation between resistance phenotype and the *kdr* genotype was investigated by calculating the odds ratio, using the epiR package in R version 4.1.1 (https://cran.r-project.org/bin/windows/base/, accessed on 5 December 2021).

The results of the WHO tubes bioassay were interpreted as follows: mortality between 98–100% = susceptible; resistance is suggested for 90–97% mortality, while resistance is confirmed for less than 90% mortality. Mortalities in the controls of more than 5% were corrected using the Abbott’s formula [68]. The unpaired Students t-test was used to compute statistical differences in mortalities using the replicate of mean mortalities for each insecticide between the three years.

## 3. Results

### 3.1. Morphological and Molecular Identification of Aedes Mosquitoes

Morphological identification established all the Aedes mosquitoes collected belonging to Aegypti species. The head region has white flat-scale patches present on clypeus and the vertex with little erect forked scales present only on the occiput. The proboscis was purely dark with no trace of the white band. The legs presented with a white stripe on femoral knee-spot and anterior portion of a mid-femur, and the thorax. The scutum was black with a pair of sub-median longitudinal white stripes.

### 3.2. Insecticide Susceptibility Tests

#### 3.2.1. Larval Bioassays

The larval bioassays conducted with the 2018 populations revealed no resistance to temephos, with LC_50_ of 0.027 mg/L (95% CI: 0.008–0.092), which is only 1.125 times higher than what was known for the laboratory susceptible colony, New Orleans (LC_50_ = 0.024 mg/L); leading to an estimated resistance ratio (RR) of 1.125 (Table 1). Likewise, the 2019 population were highly susceptible to the temephos with LC_50_ of 0.008 mg/L (95% CI: 0.005–0.013), which is even lower than the LC_50_ of the New Orleans, leading to the RR of less than 1 (RR = 0.333). However, in 2020 there was slight decrease in susceptibility to temephos with LC_50_ of 0.037 mg/L (95% CI: 0.006–0.219) which led to an increase in the RR to 1.54 when compared with the value from New Orleans.

From the larval bioassay with deltamethrin only the 2018 populations showed susceptibility (LC_50_ = 0.018 mg/L, 95% CI: 0.007–0.046, and RR = 2.250). Between 2019 and 2020, deltamethrin resistance dramatically increased, with the 2019 population exhibiting LC_50_ of 0.100 mg/L (0.030–0.356), which is ~6 fold higher than 2018 and RR of 12.5. By 2020, resistance had increased ~11 fold compared to 2018, with RR of 24.750 and LC_50_ of 0.198 mg/L (0.111–0.367) (Table 1).

Details of the resistance pattern, including LC_25_, LC_90_, and percentage mortalities for all the 7 concentrations of each of the two insecticides are provided in Appendix A. However, the average percentage mortalities with the median concentration (0.05 mg/L) of temephos was 72.2% for the 2018 collection. The resistance level with the same median concentration of the temephos increased, with mortalities reducing to 62.3% in 2019. A significant decrease in mortalities was observed in 2020 (51.9%, *p* = 0.02, t = 3.08, df = 6, for 2018 vs. 2020) (Figure 1). For 0.05 mg/L deltamethrin, a successive decrease in mortality was also observed, from 50% in 2018, to 44.9% and 34.2% for 2019 and 2020, respectively. Significant difference was obtained only when comparing mortalities between 2018 and 2020 populations (*p* = 0.04, t = 2.6, df = 6).

#### 3.2.2. Insecticide Resistance Profile of the Adult *Ae. aegypti* Population

A total of 2700 adult female *Ae. aegypti* from BUK were tested against four classes of public health insecticides (number of mosquitoes tested for each insecticide provided in Appendix A). The populations were highly resistant to both type I and type II pyrethroids (except cyfluthrin), propoxur and DDT (highest resistance) with increase in resistance observed across the three years. For example, an increased permethrin resistance was observed with 24 h mortalities of 83.8%, 74.0% and 71.0% for 2018, 2019 and 2020, respectively (Figure 2a–c). Similarly, the same pattern was observed in mosquitoes tested with ƛ-cyhalothrin, with mortality significantly reducing from 74.4 % in 2018, to 59.0 % in 2019 (*p* = 0.03, t = 2.88, df = 1) and 52.0% in 2020 (*p* = 0.008, t = 4.32, df = 1). Bioassay with deltamethrin revealed a contrasting pattern with mortality of 83.4% for 2018, which decreased to ~78% for the 2019 and 2020 populations. However, moderate resistance was initially observed with cyfluthrin bioassay in 2018 population, with an average percentage mortality of 96.7%, which reduced to 90% in 2019.

Extremely high resistance was observed toward DDT across the three consecutive years with mortality of only 2.5% for 2018 populations and 3% for 2019 and 2020, respectively. However, the Pantami population revealed a contrasting pattern, with the DDT killing 93% of the mosquitoes (Figure 2d). Details of replicates and number of females used for each insecticide, as well as mortalities with the lower concentrations are provided in Appendix A.

The knockdown rates with pyrethroids increased with time, with the highest seen with cyfluthrin and deltamethrin and the lowest from permethrin and ƛ-cyhalothrin (Appendix A). For all the pyrethroids tested, there were successive decreases in knockdown rates across the three years. For the organochlorides, over the years DDT did not inflict any knockdown in the BUK population (Appendix A), while a very high knockdown rate was observed with the Pantami population (Appendix A). Contrary to the outcome with the DDT, dieldrin showed a very high knockdown rate in BUK populations for all the three years. The BUK populations were susceptible to fenitrothion in 2018, with mortality of 98.6% and 100% in 2019. However, the 2020 population showed moderate resistance with mortalities of 93% (Figure 2a–c).

#### 3.2.3. Determination of Resistance Intensity

Time-course (gradually increasing the DDT exposure times from 60 min, through to 540 min) quantitative bioassays were conducted each year to further confirm the degree of resistance, using WHO tube bioassays (Figure 3). The lethal time that killed 50% of the populations (LT_50_) for 2018 was estimated as 352.87 min (95% CI: 359.29–376.76), with the LT_50_ gradually increasing to 369.19 min (CI: 304.87–477.60) in 2019 and 406.94 min (CI: 372.59–575.15) in 2020.

#### 3.2.4. Determination of the Role of Metabolic Resistance Enzymes Using Synergist Bioassays

Synergist bioassays were conducted to determine the contribution of metabolic enzyme systems in resistance to DDT and λ-cyhalothrin. Pre-exposure to 8% DEM for 1 h followed by 4% DDT recovered susceptibility significantly, with mortalities increasing 15-fold, from 2.5% with DDT only to 36.3% in synergised cohort (DEM plus DDT, x^2^ = 35.0548, *p* < 0.00001) in 2018 BUK population (Figure 4a). In 2019, susceptibility was restored, with mortalities increasing from ~3% with DDT only, to 28% when pre-exposed to DEM (x^2^ = 23.8595, *p* < 0.00001) (Figure 4b).

In 2020, mortalities in DEM synergised mosquitoes increased 10-fold, from only 3% with DDT to 30% with synergist (x^2^ = 26.4562, *p* < 0.00001) (Figure 4c). No mortality was recorded with controls—both the unexposed and mosquitoes exposed to only DEM.

The PBO pre-exposure significantly restored ƛ-cyhalothrin susceptibility in the 2018 BUK population, with mortalities increasing from 74.2% in the conventional bioassays with the ƛ-cyhalothrin alone, to 100% in synergised females (x^2^ = 25.1789, *p* < 0.0001) (Figure 4a). Likewise in the subsequent years, significant associations were observed between recovery of susceptibility and PBO preexposure, with mortalities increasing from 59% to 98.8% (x^2^ = 45.8409, *p* < 0.0001) in 2019 and from 52% to 98% (x^2^ = 54.000, *p* < 0.0001) in 2020 (Figure 4b,c).

### 3.3. Determination of Insecticidal Efficacy of PermaNet^®^ 3.0

Cone bioassays were conducted to investigate the efficacy of the deltamethrin-impregnated LLIN, PermaNet^®^ 3.0. The effectiveness of both the side panels (deltamethrin only) and the roof (PBO + deltamethrin) were assessed each year, with the female *Aedes aegypti* from BUK. Resistance was observed to increase temporally with the side panels, with 1 h average percentage knockdown and 24 h mortality of 50% and 94%, respectively for 2018, 42.4% and 66.4% for 2019, and 40.4% and 73.6% for 2020 (Figure 5a–c).

In contrast, exposure to the roof of the net increased the knockdown significantly and restored full susceptibility, with 100% mortality obtained at 24 h, for each experiment carried out in the three years.

### 3.4. Investigating the Role of the F1534C and V1016G kdr Mutations in Resistance

The impact of the VGSC F1534C and V1016G mutations on pyrethroid and DDT resistance was investigated by successfully genotyping contrasting phenotype females (42 each of ƛ-cyhalothrin-alive and -dead females, as well as 34 DDT-alive and 37 -dead). Lower frequency of the 1534C *kdr* mutation was observed in the ƛ-cyhalothrin -alive mosquitoes, with only 3 females homozygote resistant (7.14%, CC), 10 heterozygotes (23.81%, FC) and 29 homozygotes susceptible (69.05%, FF), compared to the dead females [6 (14.29%, CC), 13 (30.95%, CF) and 23 (54.76, FF)] (Table 2, representative gel micrographs shown in Appendix A). The 1534C *kdr* frequency was 0.309 in the alive females and 0.45 in the dead, and overall, for both 0.38. No significant correlation was obtained between the *kdr* genotype and ƛ-cyhalothrin resistance phenotype in all comparisons, with odds ratio, OR of 0.40 (95% CI: 0.09–1.76) for CC vs. FF (x^2^ = 0.78, *p* = 0.37), OR of 0.61 (0.23–1.64) for FC vs. FF (x^2^ = 0.97 (*p* = 0.33), and OR of 0.54 (0.22–1.33) for CC + CF vs. FF (x^2^ = 1.82, *p* = 0.18).

For DDT, a contrast was observed in the phenotype distribution of 1534C frequencies, with no homozygote resistant, RR in alive females (0% CC), while 12 females were heterozygotes (35.29%, FC) (Table 2, Appendix A). This is in contrast with the dead females, in which 10 individuals were homozygote resistant (27.03%, CC) and none was heterozygote (0%, FC). The 1534C *kdr* frequency was 0.35 for the alive females and 0.27 in the dead, and overall, for both, 0.31. While no significant correlation was obtained between the *kdr* genotype and DDT resistance phenotype for the CC vs. FF [OR = 0.12 (0.01–1.03) x^2^ = 3.48, *p* = 0.06] and CC + FC vs. FF [OR = 1.47 (0.54–4.05), x^2^ = 0.57, *p* = 0.45], a correlation was obtained when comparing heterozygote resistant females, FC with the homozygote susceptible, FF [OR = 14.73 (10.77–22.23), x^2^ = 9.33, *p* = 0.02].

The samples genotyped for the F1534C *kdr* mutation were also used to genotype the V1016G mutation. All the samples tested showed a band size of ~60 bp, confirming the absence of 1016G mutation (representative gel micrographs shown in Appendix A).

## 4. Discussion

This study investigated insecticide resistance in the immature stages and adults of the arboviral vector, *Ae. aegypti,* by collection of larvae and pupae in the rainy season over three years and testing them with public health insecticides, synergists, as well as an LLIN, PermaNet^®^ 3.0. The study also explored the possible mechanisms of pyrethroid and DDT resistance by genotyping resistant and susceptible females for the presence of F1534C and V1016G knockdown resistance (*kdr*) mutations in the VGSC and assessing correlation with the observed resistance to the pyrethroids and DDT.

The study revealed that *Ae. aegypti* is probably the major *Aedes* species in northern Nigeria. Indeed, this species is known to be the most invasive species in many parts of the world [1,69,70]. Several findings from African and Asian countries reported similar pattern with *Ae. aegypti* being the dominant species within the cities and *Ae*. *albopictus* dominating in the suburban areas [48,71,72]. However, Kamgang and colleagues [48] had reported an incident in which the invasive *Ae*. *albopictus* was gradually replacing the native *Ae. aegypti* in the Republic of the Congo. The invasion of this non-native species which includes its introduction, establishment, and spread posed a great challenge to ecosystems and most importantly human activities and health [70]. This is because this species together with *Ae. albopictus* are the major vector for most arboviruses in Africa, and the world at large [73]. The invasions of this species also serve as an indicator for climatic changes in many tropical and subtropical regions of the world and environmental changes in response to socio-economic development [74].

The observed temporal decreases in temephos susceptibility in less than three years suggest the likelihood of temephos resistance becoming established in the nearest future. The escalated deltamethrin resistance in the BUK larvae is of grave concern to the control and management of resistance in *Aedes aegypti* in northern Nigeria, especially when the 2018 population was susceptible, but resistance crept in, increasing 11-fold by 2020. Several reports support the findings of this research regarding the high degree of resistance to both temephos and deltamethrin. For example, Bellinato [75] and Goindin [76] reported a similar scenario where the Aedes were resistant to deltamethrin and the temephos. However, contrasting reports from Burkina Faso and Cameroon showed full susceptibility of temephos against the immature stages of *Ae. aegypti* [44,77].

The finding of multiple resistance in BUK populations is not surprising, as several studies from southern Nigeria have documented similar trends. For example, a recent study on *Ae. aegypti* from several localities within Lagos state, south-western Nigeria, has documented DDT and permethrin resistance [39] with percentage mortalities of 20–40% and 29–70%, respectively. Another study from Awka, south-eastern Nigeria (Anambra state), had documented *Ae. aegypti* as highly resistant to DDT (mortality of <10%), moderately susceptible to propoxur (mortality of 97.3%), but fully susceptible to deltamethrin (mortality of 1%) [50]. The same study described *Ae. albopictus* as resistant to DDT (mortality of 62%) and deltamethrin (mortality of 93.6%), while being fully susceptible to propoxur. The extremely high DDT resistance in BUK adult *Aedes* is in agreement with the findings of another study, carried out in Abia state in south-eastern Nigeria [78], where the *Ae. aegypti* populations were reported to be susceptible to all insecticides tested (bendiocarb, pirimiphos-methyl and deltamethrin) except for DDT (where mortality was 62.85%). These suggests a contrasting pattern of DDT resistance across Nigeria, which is of great concern. However, the level of DDT resistance seen in the Pantami population is closer to the observation of Ukpai and Ekedo above, which underscore the need to assess the resistance profile of as many *Aedes* populations as possible before programmatic decisions. A similar finding was observed in the Malaysian populations of *Ae. aegypti* [31] which were highly resistance to DDT across the country. The observation of contrasting susceptibility to pyrethroids of similar types, e.g., resistance to deltamethrin and ƛ-cyhalothrin, but susceptibility to cyfluthrin, as well as the extremely high resistance to DDT, but susceptibility to dieldrin, highlighted the complex nature of metabolic resistance. The significant recovery of DDT and ƛ-cyhalothrin susceptibility following the synergist bioassays suggests the preeminent role of metabolic resistance in the *Ae. aegypti* population from Kano. Similar findings were reported by a recent study in four locations within Lagos state [39] with DDT resistance significantly recovered following preexposure to piperonylbutoxide (mortalities in DDT conventional bioassay = 20–26%, while PBO + DDT = 75–100%) and permethrin conventional bioassay mortalities = 29–70%, while PBO + permethrin = 78–99%).

Little is known in Nigeria of the effectiveness of LLINs on non-target mosquitoes, such as Aedes. Indeed, Aedes species may shift in biting behaviour [79] from diurnal to nocturnal [80]. We have been collecting blood fed, female Aedes resting indoor (though in low densities) in several localities, while doing indoor collection of blood fed Anopheles —which suggest that this species maybe biting and resting indoors in north-western Nigeria. Indeed, several studies have documented diurnally active *Ae. aegypti* populations and in 2009 the WHO had encouraged the use of insecticide-treated materials for control of diurnally active *Ae. aegypti* [65]. For the three consecutive years (2018, 2019 and 2020), we found that the *Ae. aegypti* population were fully susceptible to the roof of PermaNet^®^ 3.0 while the side PermaNet^®^ 3.0 showed a decrease in mortality by 20.3% within three years. Herrera-Bojórquez and colleagues [64] reported the effectiveness of LLINs with declining physical and chemical integrity on *Ae*. *aegypti,* in which despite the loss of physical and chemical integrity, the left-over chemical effect still played a vital role in killing and/or repelling the mosquitoes.

The F1534C and V1016G mutations in the voltage-gated sodium channel VGSC (target site for pyrethroids and DDT) are commonly found in *Ae. aegypti* and are usually associated with insecticide resistance worldwide [81]. The mutation has been reported from many African countries including Cameroon [44], Burkina Faso and Ghana [77,82]. This study is possibly one of the first to report the presence of the F1534C mutation and its contribution to insecticide resistance in Ae. aegypti population from Nigeria. The V1016G mutation was not observed in this population, which was not surprising, as several studies have established the frequency of the F1534C tending to be higher than that of V1016G, though additive effect on pyrethroid resistance is seen in individuals carrying the two *kdr* alleles [32,36]. The absence of the V1016G and the low frequency of the 1534C resistance allele suggest the minor role of target-site insensitivity in the BUK population, compared with the more complex metabolic mechanisms. However, the observed correlation between DDT resistance and heterozygote genotype (FC) for the 1534 *kdr* mutation suggests that this mutation confers resistance, with possibly fitness cost associated with being homozygote (no homozygote resistant individual was observed among DDT-alive females and no heterozygotes among the dead). Indeed, Intan and colleagues had described a significant association between the presence of the 1534C allele and pyrethroids resistance in *Ae. aegypti* population from Malaysia. Another study from Burkina Faso by Sombié and colleagues [82] reported extremely high frequencies of F1534C (97%) mutation and established that presence of the *kdr*-mutation at the dual locus was linked to pyrethroid resistance.

## 5. Conclusions

This study established the presence of *Ae. aegypti* in northern Nigeria, confirmed multiple resistance, increasingly reported across Nigeria, in this major arboviral vector, and explored mechanisms underlying the resistance. Temporal increase in resistance was observed in both larvae and adult bioassays, with highest resistance seen toward λ-cyhalothrin in larvae and DDT in adults. The most susceptibility was observed from cyfluthrin, dieldrin and fenitrothion suggesting these compounds as potential insecticides for IRS. In addition, resistance was observed toward the conventional LLIN containing deltamethrin, but with recovery from combination LLIN containing PBO, which suggests the use of combination LLINs could provide better indoor protection from this vector. Future studies should explore more sites across Nigeria to confirm the temporal and spatial distribution of this vector and its resistance profiles. Also, transcriptomic analysis could provide in-depth information on the metabolic mechanisms operating in the field population of this important vector, to guide the stakeholders and relevant authorities in Nigeria on implementation of evidence-based control and resistance management.

## Figures and Tables

**Figure 1 insects-13-00187-f001:**
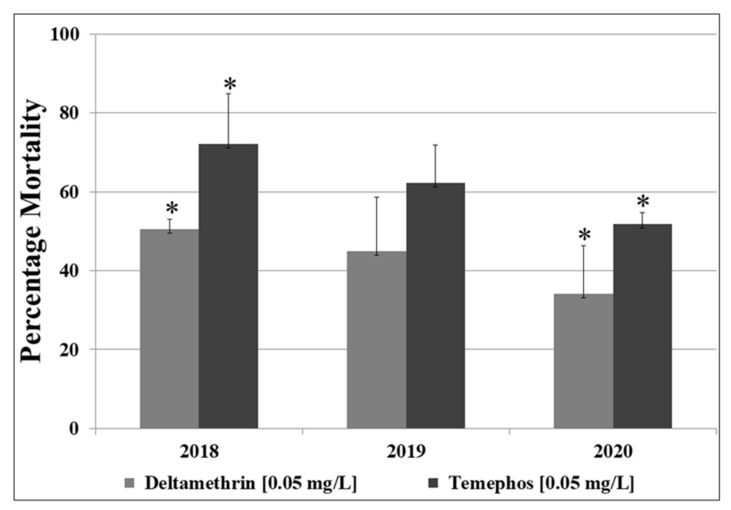
Results of larval bioassays with temephos and deltamethrin for 2018, 2019 and 2020 populations of *Ae. aegypti,* from BUK. Results are average of mortalities from four replicates for 0.05 mg/L concentration after 24 h exposure. Error bars indicate standard deviation of means. * = mortalities significantly different when comparing 2018 populations with 2020.

**Figure 2 insects-13-00187-f002:**
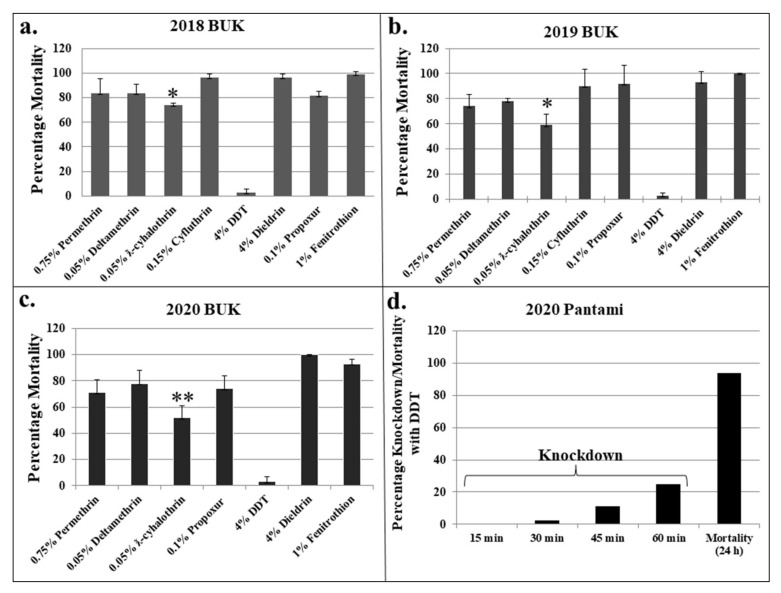
Results of WHO tubes bioassays with different public health insecticides. Results are average of percentage mortalities at 24 h post-exposure, with error bars denoting standard deviation. (**a**–**c**) indicate result from the BUK population in 2018, 2019 and 2020 respectively, while (**d**) indicates the results of 2020 Pantami population with 4 % DDT. * and ** = mortalities significantly different for ƛ-cyhalothrin when comparing 2018 vs. 2019 (*p* < 0.05) and 2018 vs. 2020 (*p* < 0.01).

**Figure 3 insects-13-00187-f003:**
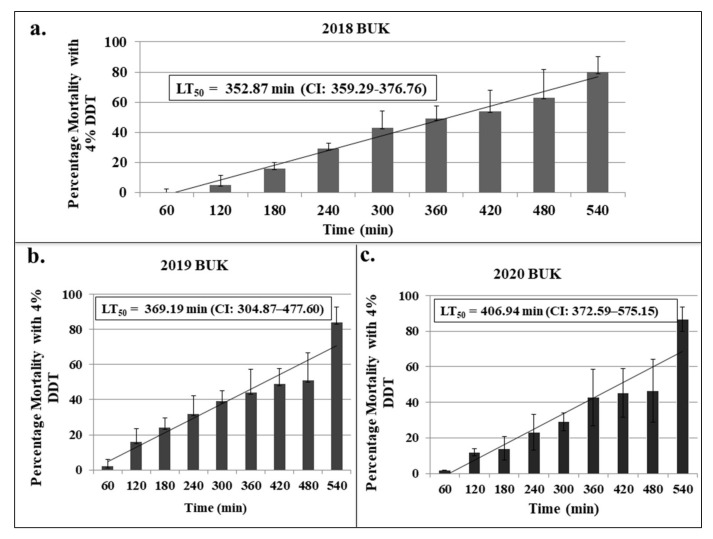
Results of WHO tubes time-course bioassays with 4% DDT. Results are average of percentage mortalities at 24 h post-exposure, with error bars denoting standard deviation. (**a**–**c**) indicate result from the BUK population in 2018, 2019 and 2020, respectively.

**Figure 4 insects-13-00187-f004:**
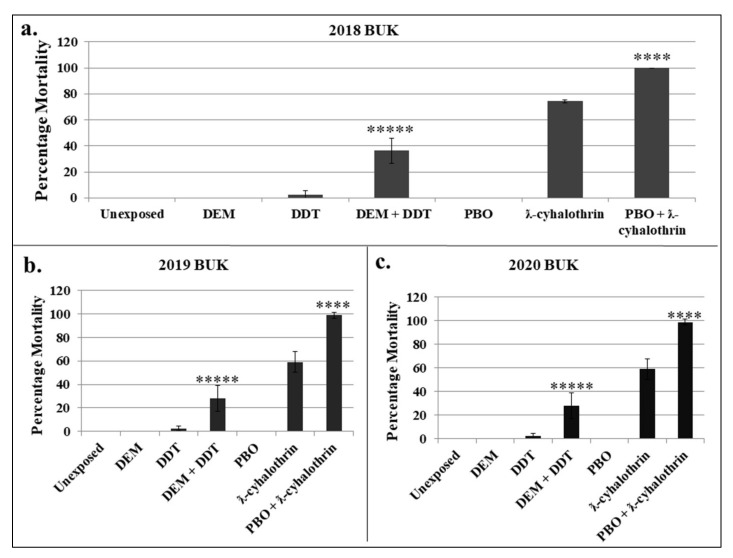
Investigation of the role of metabolic enzymes in insecticide resistance. (**a**–**c**). Effect of DEM and PBO synergists preexposure on susceptibility to DDT and ƛ-cyhalothrin, respectively, on BUK populations from 2018, 2019 and 2020. Results are average of percentage mortality for synergised and un-synergised females, with error bars indicating standard deviation. **** and ***** = significantly different at *p* < 0.0001 and *p* < 0.00001, respectively.

**Figure 5 insects-13-00187-f005:**
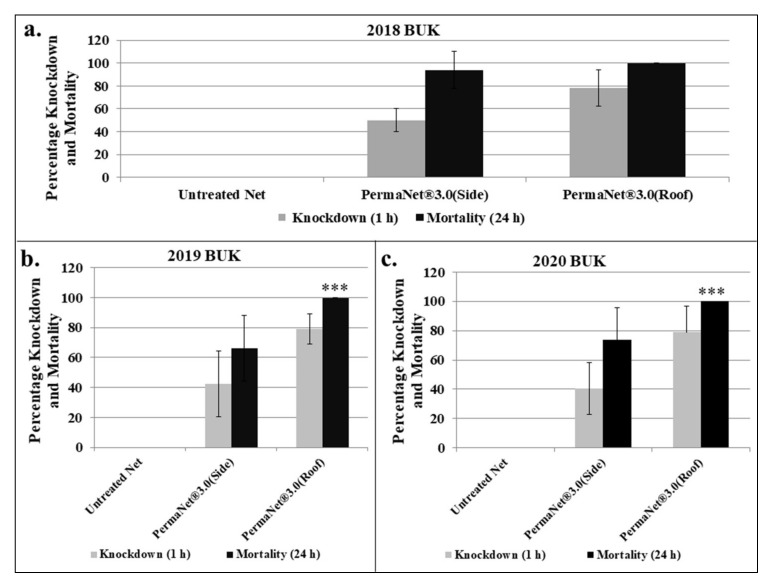
Results of WHO cone bioassays with side panels and roof of PermaNet^®^ 3.0. (**a**–**c**) for 2018, 2019 and 2020, respectively. Results are percentage averages of 1 h knockdown or 24 h postexposure percentage mortalities, with error bars indicating standard deviation. (*** = *p* < 0.001 for 2019 and 2020, respectively).

**Table 1 insects-13-00187-t001:** Temephos and deltamethrin sensitivity of BUK *Ae. aegypti* larvae.

Insecticide	Collection Year	*n*	Slope (±SE)	LC_50_ (mg/L) (95% CI)	x^2^	*p*	RR
Temephos	2018	560	0.641 (0.039)	0.027 (0.008–0.092)	21.38	0.001	1.125
	2019	560	0.582 (0.037)	0.008 (0.005–0.013)	1.53	0.009	0.333
	2020	560	0.475 (0.032)	0.037 (0.006–0.219)	29.16	0.0001	1.542
Deltamethrin	2018	560	0.588 (0.037)	0.018 (0.007–0.046)	11.99	0.035	2.25
	2019	560	0.505 (0.033)	0.100 (0.030–0.356)	16.06	0.007	12.5
	2020	560	0.416 (0.030)	0.198 (0.111– 0.367)	7.24	0.020	24.75

*n* = number of larvae used per experiment; LC_50_ = lethal concentration that killed 50% of the larvae; SE = standard error of mean; CI = confidence interval; x^2^ = Pearson Goodness-of-Fit Chi Square test; *p* = level of significance value; and RR = resistance ratio, a ratio of the LC_50_ of resistant field population compared with the LC_50_ of the fully susceptible laboratory colony (New Orleans).

**Table 2 insects-13-00187-t002:** Correlation between the 1014C allele frequency and resistance λ-cyhalothrin and DDT resistance phenotype in BUK *Aedes aegypti* populations.

Insecticides	Phenotype	*n*	F1014C Alleles			% *kdr* Frequency (RR + RS)	*kdr* Allele	Odds Ratio (RS vs. SS)	x^2^ (*p* Value)
			CC (RR)	FC (RS)	FF (SS)				
λ-cyhalothrin	Alive	42	3 (7.14%)	10 (23.81%)	29 (69.05%)	30.95	0.31		
	Dead	42	6 (14.29%)	13 (30.95%)	23 (54.76%)	45.24	0.45		
	Total	84	9 (10.71%)	23 (27.38%)	52 (61.90%)	38.09	0.38		
DDT	Alive	34	0 (0%)	12 (35.29%)	22 (64.71%)	35.29	0.35	14.73 (10.77–22.23)	9.33 (0.02)
	Dead	37	10 (27.03%)	0 (0%)	27 (72.98%)	27.03	0.27		
	Total	71	10 (14.08%)	12 (16.90%)	49 (69.01%)	30.98	0.31		

*n* = number of successfully genotyped individuals. Numbers in brackets represent percentage frequencies. Homozygote resistant alleles (RR, CC); heterozygote resistant (RS, FC); and homozygote susceptible (SS, FF).

## Data Availability

All data generated in this research is contained within the article or Appendix A provided.

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
