# Peer review of "Temporal Evaluation of Insecticide Resistance in Populations of the Major Arboviral Vector Aedes Aegypti from Northern Nigeria"

_insects, 2022, doi:10.3390/insects13020187_

Round 1
Reviewer 1 Report
The manuscript submitted by Mukhtar and Ibrahim speaks about temporal escalation of insecticide resistance in populations of the major arboviral mosquito vector Aedes aegypti from northern Nigeria. The manuscript is well written and the study design is acceptable. However, some concerns weaken the quality of the manuscript
1- Line 11: Authors cited both Aedes aegyti and Ae. albopictus as Asian tiger mosquitoes. I don’t that this is true for Ae. aegypti. Indeed, the origin of this species is not Asia continent but Africa. This species migrated out of Africa in the favor of slave trade from Africa to other areas.
2-Line 49: Authors cited West Nile viruses among diseases for which Ae. aegypti is an important vector. I don’t think this vector is so important in the transmission of this disease. Indeed, even if the virus had been isolated in Aedes aegypti, its transmission by this mosquito is not so important in the literature. So I would like to suggest the authors to be prudent when talking about that.
3- The 4 first paragraphs of the introduction make this latter to be too long. This first part is also a bit far away with the topic of the paper. I understand that authors wanted to present the importance of diseases transmitted by Aedes mosquitoes, but I fill the text is enough. I would have been great to talk a bit about the current situation worldwide with emphasis on the current situation of emergence of these diseases in in Africa during the two last decades. After that talk about the need for Africa to be start preparing now to be ready to control coming outbreaks. Because the control is mainly based on insecticide based vector control, is important to know the insecticide susceptible profile of the populations. Presented a bit like that, the introduction would has been easy to read and probably not so long.
4- Line 74 seems to have no sense there. For me there is no link with the phrase before and the one coming after. May be it should be removed.
5- Line 176-201: “Mosquito populations identification to the species level”. I wonder why author performed molecular analyses to identify Aedes mosquito. In my knowledge these species are not sibling ones and are easy to identify morphologically using existing identification keys. This was clearly explained and demonstrated by the authors in the results section (lines: 317-321). It’s not clear to understand why performing molecular analyses if this does not bring any plus. It’s like authors only need to have more results in their manuscript. This part of the work could be remove without changing the results of the identification. We should always think about the reproducibility of a protocol described in one article. One researcher could want to reproduce this work in another context, so we should avoid making he performing plenty analyses with no scientific interest. It’s not because you have means and condition to perform all molecular analyses you want that we can run anyone with no pertinent scientific interest.
6- Line 202 is repeated
7- Larval bioassays: The origin of larvae used for bioassays is not clearly indicated. Indeed, one can think that larvae used here are the ones collected during field collection. If is the case I don’t think is the go way to do. Please clearly state where larvae used for these bioassays come from.
8- Lines 218-223: I think this should go in the data analysis section. Furthermore it looks like redundancy with lines 301 -302.
9- Lines 240 – 244: What do authors understand by “adequate information….”? Stated like this this justification in highly weak to explain why they decide to modify a recommended protocol for exposure time to characterize insecticide resistance profile in a population. More needed to be said to prove the pertinence of changing the exposure time. If not, it will look like authors just need to have plenty results in their manuscript.
10- Lines 259 – 269: Determination of efficacy of LLIN on Aedes mosquitoes.
What was the purpose of this analysis? This is not clearly justified in the manuscript. In Africa aedes mosquitoes are mostly known as exophilic and exophagic, even if some isolated individual are sometime found indoor. May be the authors could more justify this analysis.
11- Figure 1: it looks like there is a confusion between 2019 and 2020 for the results of with deltamethrin. Please could you double-check?
12- On the figure 2-d it’s indicated 2020 whereas in the legend picture d indicate 2018 results for Pantami. Could you verify this?
13- Lines 414 – 420: Usually according to WHO standard, when talking about “resistance intensity”, this is related to insecticide doses. Please could you clearly explain how the time of exposure could be informative on the resistance intensity? Furthermore, do the difference you observed between the 3 years were significantly different?
14- Figures 1, 2, and 3: I suggest you put the max value of the mortality at 100% instead of 120%
15- Lines 574 -577: This looks very speculative. Please bring more accurate information that can hardly justify this analysis. It’s not just because you collected some specimen during your Anopheles mosquito collection that it looks normal to spend time and money to run experiments on a species which is mostly exophilic and exophagic. What is the scientific purpose of that?
Author Response
Dear Reviewer,
We have attempted to address the corrections and the suggestions you made on our mansucript (both in the cleaned and the tracked version). Please find our responses below in form of word document.
Kind regards.

Reviewer 2 Report
#Manuscript ID: insects-1543480
In the paper entitled `Temporal escalation of insecticide resistance in populations of the major arboviral mosquito vector Aedes aegypti from northern Nigeria`, Mukhtar and Ibrahim described the possible resistance of Ae. aegypti to several insecticides that can help public health agencies in pest control programs. The paper is well written and sounds scientific. Yet, I have comments on data analyses and presentations that need to be clarified and revisited to support the author’s claims.
(i) In figures 1 and 2, the authors presented data on insects’ mortality from three years of collection. But no analysis was provided to compare the mortalities rates among the years. Otherwise, the authors only report means and SD and, in the results, claim differences among years as noted in lines 362-368 (Figure 1) and 378-388 (Figure 2a-c). Thus, I suggest the authors: (i) perform an analysis to compare mortality among years for the same insecticide; (ii) represent data for each insecticide comparing the years. In the current form, it is not clear to see the variation by year for the same molecule as reported by the authors in the result.
(ii) Figure 2d can be presented alone since it is a different result apart from a, b and c.
(iii) It is unclear how the authors extracted the mean lethal time since it was estimated from different insects exposed to the other times to the insecticide using a linear analysis. Usually, lethal times are estimated by survival probability analysis through, for instance, using Kaplan Meier estimators to get lethal times. For example, see the references:
Miranda et al., 2022. Exposure to copper sulfate impairs survival, post-embryonic midgut development and reproduction in Aedes aegypti. Infection, Genetics and Evolution. https://doi.org/10.1016/j.meegid.2021.105185
Silva et al., 2019. Exposure of mosquito (Aedes aegypti) larvae to the water extract and lectin-rich fraction of Moringa oleifera seeds impairs their development and future fecundity. Ecotoxicology and Environmental Safety. https://doi.org/10.1016/j.ecoenv.2019.109583
(iv) It is unclear in figure 4 which comparison the authors refer to here. Please clarify with an appropriate analysis for multiple means.
(v) In all bioassays, the authors do not precisely report the number of replicates used. On the other hand, we read ‘minimum of four replicates’ (Line 229). Please provide the precise number of replicates used.
Minor points
Lines 103-150: Please split this paragraph up into two or three. It is not easy to follow all the information provided here.
Check references carefully are some names in italic format are missing
Author Response
Dear Reviewer,
We have attempted to correct this manuscript based on your corrections and recommendations. Please see our responses in the attached word document.
Kind regards.

Round 2
Reviewer 1 Report
NA
Author Response
Authors appreciate reviewer # 1 accepting the corrections made on this manuscript.

Reviewer 2 Report
Dear authors,
I have reviewed the manuscript's new version and found it suitable for publication. I just pointed out minor editorial reviews before publication.
L 189 and 191: Please remove (ref).
L 296: Please correct ae to Ae.
L 306: Please don’t italicize spp.
L 313: Aedes stegomyia
L 336 and 337: Please remove (ref).
L 428-430: It sounds like results instead of objectives. Please rephrase.
L 452: Change Aedes to Aedes
L 630: Please clarify resistance intensity. It is unclear what does it mean.
L 630: Please remove (ref).
L 633: Do the authors mean and?
L 654: Please remove (ref).
L Table 1: Please add P-values and Chi-square values to provide the complete information to readers.
L 1366: Please remove (ref).
Author Response
Dear Reviewer 2.
Thank you very much for the efforts to make this manuscript better. We have corrected it. Please see our responses in the attached word document.
